# Super-Resolution Multicomponent Joint-Interferometric Fabry–Perot-Based Technique

**Yu Zhang** [1,2,3], **Qunbo Lv** [1,2,3], **Yinhui Tang** [1,2,3], **Peidong He** [1,2,3], **Baoyv Zhu** [1,2,3], **Xuefu Sui** [1,2,3], **Yuanbo Yang** [1,2,3], **Yang Bai** [1,2,3] and **Yangyang Liu** [1,2,3,*]

1   Aerospace Information Research Institute, Chinese Academy of Sciences, Beijing 100094, China
2   School of Optoelectronics, University of Chinese Academy of Sciences, Beijing 100049, China
3   Department of Key Laboratory of Computational Optical Imagine Technology, Chinese Academy of Sciences, Beijing 100094, China
*   Correspondence: liuyy@aircas.ac.cn; Tel.: +86-1891-161-6008

**Abstract:** We propose a new spectral super-resolution technique combined with a Fabry–Perot interferometer (FPI) and an interferometric hyperspectral imager. To overcome the limitation of the maximal optical path difference (OPD) on the spectral resolution, the object spectrum is periodically modulated based on the FPI, and an optical Fourier transform of the modulated spectrum information is performed using a double-beam interferometer to obtain an interferogram. Drawing on the concept of nonlinear structured light microscopy, the displacement of the high-frequency interference information in the interferogram after adding the FPI is analyzed to restore the high-frequency interference information and improve the spectral resolution. The optical system has a compact structure with little impact on complexity, spectral range, or luminous flux. Our simulation results show that this method can realize multicomponent joint-interference imaging to obtain spectral super-resolution information. The effects of the FPI's reflectance and interval are analyzed, and the reflectance needs to be within 20~80% and the interval must be as close as possible to the maximum optical range of the interferometer. Compared with previous, related innovations, this innovation has the advantages of higher system stability, higher data utilization, and better suitability for interferometric imaging spectrometers.

**Keywords:** interferometric imaging; Fabry–Perot interferometer; joint-interference of multi-split components

## 1. Introduction

Interferometric spectral imaging is based on the double-beam interference principle, where the object spectrum undergoes an optical Fourier transform on the detector to obtain an interferogram. Spectral information is obtained by the inversion of the interferogram [1]. With the development applications of spectroscopy in many fields, such as atmospheric environment monitoring, water quality monitoring, and mineral exploration, the spectral resolution of interferometric spectral imagers has become increasingly required [2–7].

The spectral resolution of an interferometer is limited by its maximum effective optical path difference (OPD), the scan step size of which determines the wavelength range of the spectrum. Interferometric spectroscopic imaging techniques can be divided into temporal, spatial, and combined (temporal and spatial) modulations according to the modulation mode. In time-modulated systems, a large-stroke reciprocating motion of the moving mirror is difficult to achieve. This is mainly due to the requirement of a highly accurate and stable control system for the moving mirror [8,9]. In addition, increasing the OPD requires higher beam collimation, which can lead to a reduction in the luminous flux of the optical system. Nonetheless, the demand for simultaneous image and spectral

information of targets in many applications has spurred research on various methods for maximizing the OPD. The main research methods for this goal include the use of the parallel or rotational motion of angular mirrors to achieve interferometric optical path folding, which accomplishes the multiplication of the maximal OPD [5,10–12]. However, this approach increases the size and complexity of the interferometric optical path and introduces errors (i.e., traverse and wobble) during the motion of the corner mirror. Moreover, as the number of optical path folds increases, higher requirements are imposed for the collimation of the beam [13–15]. All of these factors affect the interference intensity. Meanwhile, only a small number of methods have been applied to spectral imaging. Under the existing conditions of selectable interferometric components, improving the maximal OPD, and even the spectral resolution of interferometric imaging spectrometers, has remained a difficult problem to solve.

Fabry–Perot interferometers (FPIs) are based on the principle of multi-beam interference, with the advantages of a small size, high transmittance, strong spectroscopic ability, and good structural stability. These have made FPIs suitable for small and high-precision hyperspectral imaging detection fields [16–20]. Despite being a primary spectroscopic component in spectral imagers, several previous studies have demonstrated the combination of a FPI with other imaging spectrometers to improve their spectral resolution, including interferometric imaging spectrometers [21–25]. The principle of such a combination is to use an interval-tunable FPI to produce very sharp comb-like transmission spectra, while sampling each transmission peak more finely at a different location in the spectral sampling interval of the original spectral imager [21,23]. The improvement in spectral resolution achieved via this resampling method depends mainly on the width of the FPI transmission peaks. Although this method holds great potential for spectral resolution improvement, it still engenders a number of challenging problems. The method requires a FPI with a narrow spectral transmission peak width and frequent interval changes, which is demanded for its high reflectance, optical accuracy, and interval-tuning accuracy. Nevertheless, it is worth noting that certain limitations apply to these indicators. In addition, the method adjusts the wavelength of the transmission peak center by tuning the FPI interval. Thus, the wavelength interval of adjacent transmission peaks will be changed for every resampling, which is concurrently related to the field of view. This will cause more difficulties during the early stage of spectrum calibration and in later stages for the reconstruction of the spectrum. The interferometric imaging spectrometer is characterized by a high spectral resolution, wide spectral range, and indirect acquisition of spectra. Therefore, the use of this resampling method makes the above-mentioned problems more prominent.

A comparative analysis of the aforementioned two interferometer types shows that the transmission spectra of the two-beam and multi-beam interferences are periodic functions of the wave number. It follows that the combination of the two types of interference components can achieve a joint, cooperative interference in the working mechanism, herein referred to as "multicomponent joint-interference" (MJI). It uses multi-beam interference to increase the possibility of a frequency shift for double-beam interference transmission spectral information, thereby providing a new solution for improving the spectral resolution.

In this study, we propose a new multicomponent joint-interference hyperspectral imaging (MJI-HI) technique to obtain a higher spectral resolution, based on a FPI. The role of the FPI is to modulate the spectrum with spectral mixing by multi-beam interference. The resulting interferogram contains modulated frequency spectral information. Owing to the modulation effect of the FPI, high-frequency interferometric information outside the maximal OPD is encoded into low-frequency interferometric information, which can be obtained by the inversion algorithm. This method of enhancing the spectral resolution can be equated to increasing the maximum OPD of an interferometric imaging spectrometer. In this paper, we introduce this with a Michelson interferometer, which can be similarly applied to other two-beam interferometers. The principle is based on the processing

of the spectral spectrum; so, theoretically, it can also be applied to dispersive imaging spectrometers, with the caveat that the spectrum needs to be Fourier-transformed first. In this paper, the principle of MJI-HI and its feasibility are demonstrated via simulation, and the effects of the FPI reflectance and interval on super-resolution spectra are discussed and compared with the super-resolution method of resampling. Compared to a single dual-beam interferometer, our proposed MJI-HI technique can achieve a twofold higher spectral resolution, while being simple, compact, and easy to set up. In addition, it has better stability without increasing the interferometric optical path, hence affecting the luminous flux less. The reflectance and interval of the FPI are the most important indicators in this study. The interval determines the resolution of super-resolution spectra, while the reflectance mainly affects the accuracy of super-resolution spectra, which have been chosen in the range of about 20% to 80%. Moreover, compared to the resampling method, our proposed technology has higher system stability, 7~35 times more-efficient data usage at the same spectral resolution, lower requirements for the FPI index, lower difficulty in spectral restoration, and is more suitable for the spectral super-resolution of interferometric spectrometers.

The remainder of this paper is organized as follows: Section 2 introduces the basic principle of the MJI-HI technique. Section 3 describes the inverse calculation method of the spectrum, and Section 4 verifies the feasibility through simulation experiments, discusses the effect of the FPI's reflectivity and interval, and presents a comparative analysis with the same type of study.

## 2. Principle Analysis of MJI-HI Spectral Super-Resolution Based on a FPI

### 2.1. Principle of the FPI

The FPI consists of two parallel, semi-reflective mirrors separated by an interval $d$ with a medium refractive index $n$, as shown in Figure 1. A parallel beam of wavenumber $\nu$ is incident on the FPI, with an incident angle $\theta$. After several reflections and transmissions, the angle of the outgoing light remains unchanged. The OPD between adjacent beams is given by $\Delta_{FPI} = 2nd\cos\theta$, whereas the phase difference is $\delta_{FPI} = 4\pi nd\nu\cos\theta$.

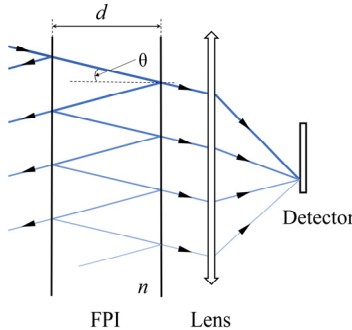

**Figure 1.** Schematic diagram of the FPI principle.

The outgoing beams of the FPI converge onto the image plane via the optical system for multi-beam interference. The effect of interference can be seen as generating a transmission spectrum $T_{FPI}$ as follows:

$$T_{FPI}\left(\delta_{FPI}\right) = \frac{\left(1-R\right)^2}{1+R^2-2R\cos\delta_{FPI}},$$

(1)

where $R$ is the reflectance of the FPI surface. The Taylor expansion of Equation (1) yields [18]

$$T_{FPI}\left(\delta_{FPI}\right)=\frac{1-R^2}{1+R^2}\left[1+\xi\cos\delta_{FPI}+\left(\xi\cos\delta_{FPI}\right)^2+\left(\xi\cos\delta_{FPI}\right)^3+...\right], \tag{2}$$

where $\xi = 2R/(1 + R^2)$. Thus, $T_{FPI}(\delta_{FPI})$ can be rewritten as follows [18]:

$$T_{FPI}\left(v\right)=\sum_{m=0}^{\infty}a_m\cos(m\delta_{FPI})=\sum_{m=0}^{\infty}a_m\cos(m\cdot4\pi ndv\cos\theta) \tag{3}$$

Thus, $T_{FPI}(v)$ is composed of a fundamental wave, with a frequency of $2nd\cos\theta$ and its higher harmonics. Because $\xi$ is only related to the FPI's reflectivity $R$, $a_m$ is also only related to $R$. From another perspective, $T_{FPI}(v)$ is a periodic function of $v$. Any periodic function can be expanded into a Fourier series, and the value of $a_m$ can be calculated directly using the Fourier series expansion formula.

### 2.2. Principle Analysis of MJI

Previous studies have found that MJI has the possibility for compatible synergistic effects. First, the transmission spectrum of the FPI is a periodic function, which means that it can be used as a modulating wave for spectral information to improve the spectral resolution [18,19]. In addition, the transmission function frequency is related to the OPD of the FPI, which can reach the same order of magnitude as the OPD of the dual-beam interferometer; thus, the frequency can meet the requirement [17,26,27].

Available knowledge indicates that the choice of different modulation principle types for the dual-beam interferometer (i.e., temporal, spatial, and joint spatial–temporal modulations) has less influence on the joint-interferometric technique based on a FPI and dual-beam interferometry. Therefore, here we use a Michelson interferometer (Figure 2) as an example to illustrate the principle underlying this technique.

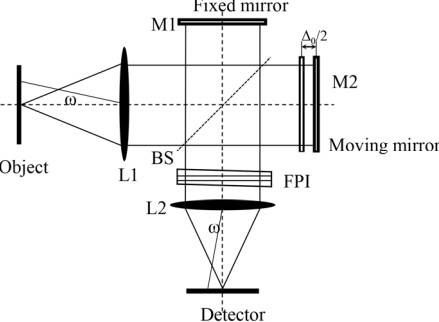

**Figure 2.** MJI-HI optical system structure.

First, the object beam is collimated by L1 into the double-beam interferometer. Then, the parallel light reflected by the fixed mirror and the moving mirror enters the FPI. The outgoing light then passes through L2 and converges at the imaging point. At this stage, the light at each image point exhibits double-beam interference and multi-beam interference. It is worth noting that the aperture size of the reflecting surfaces, on both sides of the FPI, must be fully calculated during the FPI design. The position and size of the collimated beam incident on the FPI after passing through the Michelson interferometer must be calculated based on the maximum field of view. Because the collimated beam is reflected several times inside the FPI, the apertures of both the FPI exit surface and rear optical system are larger than the incident surface. The incident and exit aperture sizes should be fully calculated during the FPI's design to ensure that multiple-beam interference can occur at the maximum field-of-view image point. The intensity of light at the image plane with wave number v is

$$I_D(\Delta,\nu)=R_{MI}T_{MI}B(\nu)T_{FPI}(\nu)\left[1+\cos(2\pi\Delta\nu)\right],\tag{4}$$

where $R_{MI}$ and $T_{MI}$ are the reflectance and transmittance of the Michelson interferometer beam splitter, respectively, and $B(\nu)$ is the spectrum of the object. Removing the DC component, the total light intensity at the image plane after MJI-HI is:

$$\begin{aligned}I(\Delta)&=\int_0^\infty B(\nu)T_{FPI}(\nu)\cos(2\pi\Delta\nu)d\nu\\&=I_0(\Delta)\otimes FT\left[T_{FPI}(\nu)\right]\end{aligned},\tag{5}$$

where $I_0(\Delta)$ is the Fourier transform of the object spectrum $B(\nu)$, and $I_0(\Delta) = FT [B(\nu)]$. The obtained interferogram $I(\Delta)$ is the convolution of the original interferogram $I_0(\Delta)$ and the Fourier spectrum $FT [T_{FPI}(v)]$ of the FPI transmission spectrum $T_{FPI}(v)$ in Equation (3). $I_0(\Delta)$ shifts in frequency during the convolution.

Structured light microscopy is a technique for the super-resolution of spatial domain information that enables a breakthrough in the optical diffraction limit [28–30]. Its illumination light is a high-spatial-frequency streak light, which is equivalent to generating a modulating wave in the frequency domain to modulate the sample spatial information. This allows high-frequency spatial information outside the cutoff frequency of the optical system to be mixed and encoded into the low-frequency-domain space within the cutoff frequency. The inversion algorithm can unmix the high-frequency information of the original frequency spectrum and expand the cutoff frequency. The maximal OPD of an interferometric imaging spectrometer is the cutoff frequency of the probe spectrum. With the modulation and unmixing approach of nonlinear structured light, it is also possible to apply modulated waves to the interferometric imaging spectral information. This enables the achievement of spectral mixing and then inversion to improve the spectral resolution.

### 2.3. Spectral Mixing of MJI

As shown in Figure 3a,b, the object spectrum is set as $B(\nu)$ and its Fourier transform as $I_0(\Delta)$, whereas $L$ is the maximum OPD of the two-beam interferometer. When $B(\nu)$ is modulated with a cosine transmission spectrum of frequency $k$, that is, $B(\nu)\cdot[1+\cos(2\pi k\nu)]/2$, the spectrum and its Fourier transform are as shown in Figure 3c,d.

$$\begin{aligned}I_k(\Delta)&=I_0(\Delta)\otimes FT\left\{\frac{1}{2}[1+\cos(2\pi k\nu)]\right\}\\&=\frac{1}{2}I_0(\Delta)+\frac{1}{4}I_0(\Delta+k)+\frac{1}{4}I_0(\Delta-k)\end{aligned}\tag{6}$$

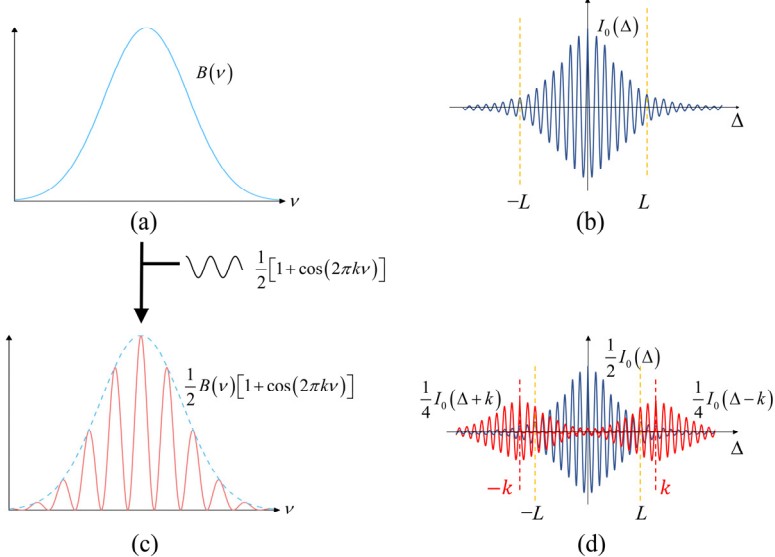

**Figure 3.** (**a**) The object spectrum; (**b**) the frequency spectrum of the object spectrum; (**c**) the spectrum after cosine modulation; (**d**) the frequency spectrum of the spectrum after modulation; the red curve represents the displacement component.

It can be observed that the effect of a single cosine-modulated wave in the frequency domain is the duplication, translation, and superposition of the frequency spectrum $I_0(\Delta)$; that is, the displacement component $I_0(\Delta \pm k)/4$ is generated. In the present study, we refer to this spectral mixing. In this process, high-frequency information is encoded into the range of invertible calculations.

When $T_{FPI}(v)$ is used as the modulation wave of spectrum $B(v)$, the modulated spectrum is as shown in Figure 4a. $T_{FPI}(v)$ can be decomposed into a fundamental wave and infinite number of higher harmonics. Each of these can be viewed as a separate cosine-modulated wave. From Equations (3) and (5), the Fourier transform of the modulated spectrum can be obtained as [28–30]:

$$I(\Delta) = I_0(\Delta) \otimes \mathrm{FT}\left[\sum_{m=0}^{\infty} a_m \cos(m \cdot 4\pi n d v)\right]$$
$$= a_0 I_0(\Delta) + \frac{1}{2}\sum_{m=1}^{\infty} a_m I_0(\Delta \pm m \cdot 2nd) \tag{7}$$

Because $a_0$ and $a_m$ are related only to $R$, they can be used as known quantities. Therefore, the effect of the FPI on the frequency spectrum $I_0(\Delta)$ is to replicate $I_0(\Delta)$ infinitely, translate them at equal intervals, and linearly superimpose them, as shown in Figure 4b.

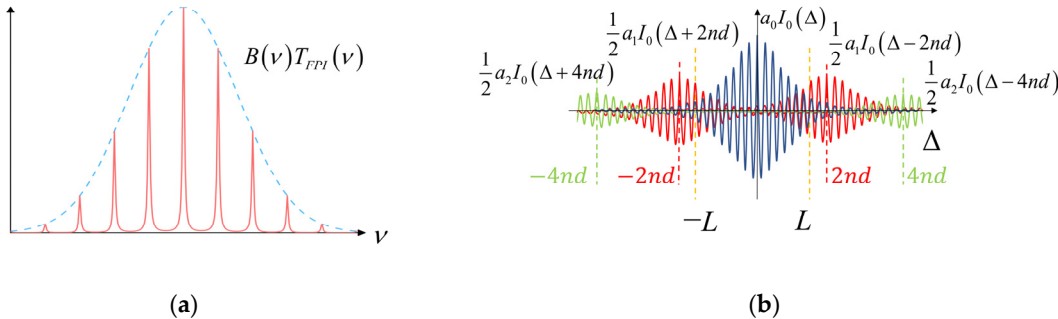

**Figure 4.** (**a**) The spectrum after FPI modulation; (**b**) the frequency spectrum of the spectrum after FPI modulation; the red curve is the m = 1 order displacement component; the green curve is the m = 2 order displacement component, and other higher order displacement components are omitted from the figure.

Thus far, by using the FPI to load high-frequency information in the spectrum into these modulated waves, each cosine-modulated wave achieves one spectral mixing. Moreover, the frequency spectrum produces displacement components $a_m I_0(\Delta \pm m \cdot 2nd)/2$. These displacement components move the otherwise unobservable high-frequency information to within $L$ and superimpose it onto the original spectrum $I_0(\Delta)$. Through the un-mixing inversion calculation, an interferogram with a wider OPD range is obtained, thereby improving the spectral resolution.

### 3. Spectral Super-Resolution Inversion for MJI-HI

Because the interferogram $I(\Delta)$, obtained by adding the FPI, is a linear combination of multiple displacement components of the original interferogram $I_0(\Delta)$, the latter cannot be calculated directly. Therefore, it is necessary to obtain multiple $I(\Delta)$ values to establish a linear system of equations for $I(\Delta)$. For this, based on Figure 2, a beam-splitter BS2 was added after the Michelson interferometer. As shown in Figure 5, one path of light behind BS2 is imaged on Detector 1 through lens L2, whereas the other is imaged on Detector 2 through the FPI and lens L3. It is worth noting that the effects produced by the reverse-transmitted beam of the FPI must be taken into consideration. The reverse-transmitted beam may go through the reflection of BS2, back into the Michelson interferometer, and then reflect back to BS2, which in turn affects Detector 1 and Detector 2. Therefore, a wedge substrate with a tiny angle is required in the FPI design to mitigate the problem of the reverse beam [21,23,24].

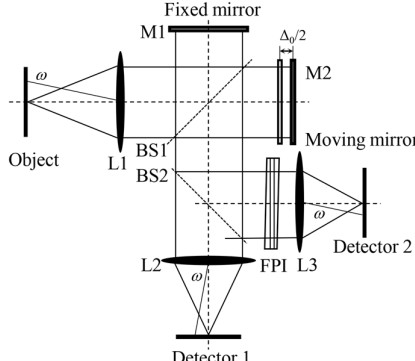

**Figure 5.** Improved MJI-HI optical system structure.

Moving M2 changes the OPD for scanning, and Detectors 1 and 2 acquire a set of interferograms. The interferogram acquired on Detector 1 is similar to that of a classical interferometric spectral imager, $I_1(\Delta) = I_0(\Delta)$, where $\Delta \in [-L, L]$. Let the FPI interval $d$ and refractive index of medium $n$ be fixed to satisfy $2nd = 2L$. The interferogram obtained for Detector 2 is then obtained, according to Equation (7) [28–30], by

$$I_2(\Delta) = a_0 I_0(\Delta) + \frac{1}{2} \sum_{m=1}^{\infty} a_m I_0(\Delta \pm 2mL), \tag{8}$$

where $\Delta \in [-L, L]$. Because the spectral energy is mainly concentrated in the low-frequency region and coefficient $a_m$ decreases with increasing $m$, the displacement component has little energy within $\Delta \in [-L, L]$. The information of the interferogram $I_0(\Delta)$ outside $\Delta \in [-2$

*L*, 2 *L*] and the higher displacement components for $m \geq 2$ are neglected. Thus, Equation (8) simplifies to

$$I_2(\Delta) = a_0 I_0(\Delta) + \frac{1}{2} a_1 I_0(\Delta + 2L) + \frac{1}{2} a_1 I_0(\Delta - 2L), \tag{9}$$

where $\Delta \in [-L, L]$. Because the interferograms $I_0(\Delta)$, $I_1(\Delta)$, and $I_2(\Delta)$ are all even functions, and $\Delta$ is discrete, Equation (9) can be expressed in the form of a linear system of equations [28–30]:

$$\begin{bmatrix} I_1(\Delta) \\ I_2(\Delta) \end{bmatrix} = \begin{bmatrix} 1 & 0 \\ a_0 & \frac{1}{2} a_1 \end{bmatrix} \begin{bmatrix} I_0(\Delta) \\ I_0(2L - \Delta) \end{bmatrix}, \tag{10}$$

where $\Delta \in [0, L]$. Since it is an even function, $I_0(\Delta)$, $\Delta \in [-2L, 2L]$ can be solved. This achieves a twofold increase in the spectral resolution. For the off-axis image point, the field-of-view angle is $\omega = \theta$ when using the FPI with a vacuum interval. In this case, $\Delta$ in Equation (10) should be changed to $\Delta \cos\theta$ and $L$ to $L\cos\theta$. The proposed spectral super-resolution method can improve the maximal OPD of the off-axis image point from $L\cos\theta$ to $2L\cos\theta$. Therefore, the spectral resolution can be improved, regardless of whether it is the central image point of the field of view. Interestingly, the interferogram inversion process is theoretically the same for the whole field of view. Compared to the resampling method, our proposed method has greater practical applicability.

The FPI does not need to satisfy $2nd = 2L$, and an FPI with a different interval can be added to the optical path corresponding to Detector 1. However, it is necessary to establish a linear equation system of $I_1(\Delta)$, $I_2(\Delta)$, $\Delta \in [-L, L]$ related to $I_0(\Delta)$, $\Delta \in [-2L, 2L]$, as shown in Equation (10), based on an appropriate simplification of $I(\Delta)$. Moreover, this method can perform multiple samplings by changing the FPI interval and establishing a higher-dimensional linear equation system to achieve numerous improvements in spectral resolution. Therefore, the proposed method exhibits substantial flexibility and scalability.

## 4. Results and Discussion

### 4.1. Simulation Conditions

In the simulation experiment, it is assumed that the $B(v)$ range of the ground object spectrum is 1.0~2.0 μm, that is, 5000~10,000 cm$^{-1}$. Let the maximal OPD of the two-beam interferometer be $L = 3$ cm; that is, the spectral resolution is 0.167 cm$^{-1}$, and the FPI satisfies $2nd = 2L$, where the OPD scan step is $l = 0.5$ μm. We consider this as an example for simulation verification. Because the input continuous spectrum cannot be used in the simulation process, the input spectrum is a discrete spectrum with an interval of 0.001 cm$^{-1}$, which is approximated as a continuous spectrum. MATLAB was used for this simulation.

As shown in Figure 6, we used three input spectra: a cosine function with spectral period frequency $\omega = 5$ cm, a Gaussian function with a half-height width of 0.075 cm$^{-1}$, and atmospheric absorption spectrum data calculated from the US standard atmosphere (1976) and the HITRAN API.

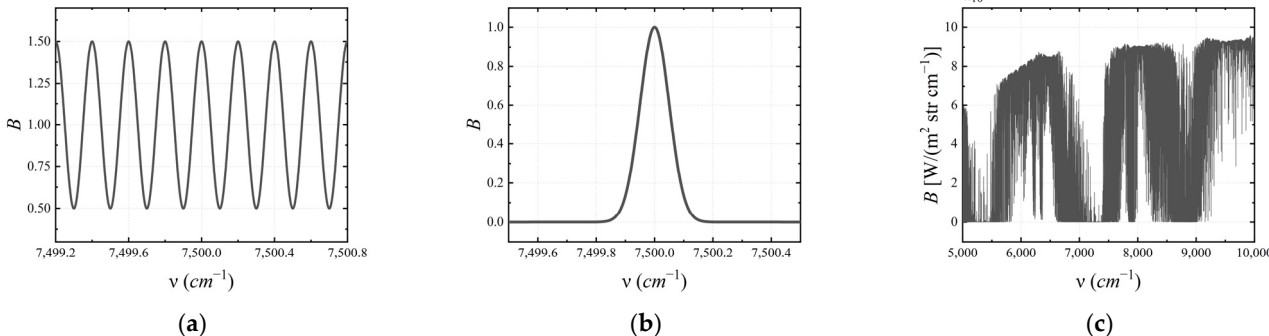

**Figure 6.** Input spectrum: (**a**) cosine function; (**b**) Gaussian function; and (**c**) spectral data.

### 4.2. Simulation Results

The ideal interferogram $I_0(\Delta)$ is obtained by directly Fourier transforming the input spectrum $B(\nu)$, and the super-resolution interferogram $I'_0(\Delta)$ is obtained by MJI-HI. Figure 7 shows a comparison diagram of $I'_0(\Delta)$ and $I_0(\Delta)$, where the latter exhibits an interference peak at $\Delta = \pm5$ cm when the input spectrum is a cosine function. Interference peak information is lost when using a classical interferometric imaging spectrometer. In contrast, $I'_0(\Delta)$ recovers the interference peak at $\Delta = \pm5$ cm, as shown in Figure 7a. An enlarged view of this part in Figure 7b clearly indicates that the result of the calculation of $I'_0(\Delta)$ has a high level of accuracy. Since the input spectral range is 1.0~2.0 μm, $I_0(\Delta)$ is here a curve similar to the Sinc function.

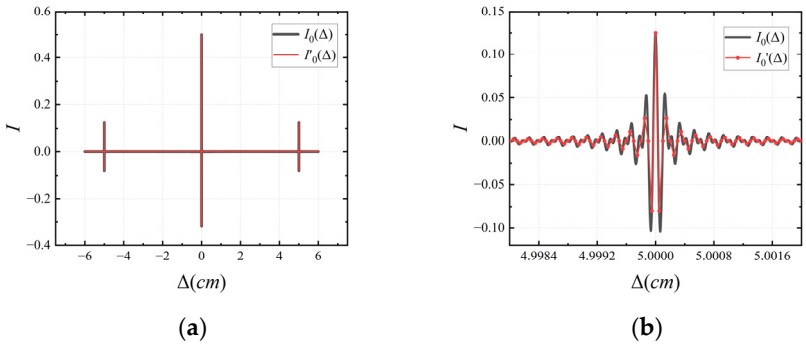

**Figure 7.** (**a**) The interferogram when the input spectrum is a cosine function; (**b**) the partial magnification comparison of this interferogram.

The inverse Fourier transform of $I'_0(\Delta)$ yields MJI super-resolution spectrum $B'(\nu)$. The Nyquist sampling theorem is satisfied by adding zeros at both ends of the interferogram. To ensure that the improvement in spectral resolution does not originate from the sampling density, we set the same amount of data as $I'_0(\Delta)$, $\Delta \in [-2L, 2L]$ by adding zeros at both ends of $I_0(\Delta)$ $\Delta \in [-L, L]$. Then, we set the same sampling density for $B'(\nu)$ and $B_{classic}(\nu)$. Figure 8 shows the comparison between $B'(\nu)$ with the input spectrum $B(\nu)$ and the low-resolution spectrum $B_{classic}(\nu)$, obtained from a classical interference imaging spectrometer with the same maximal OPD. It can be seen that $B'(\nu)$ can restore the cosine waveform well, whereas $B_{classic}(\nu)$ completely loses the high-frequency information of the spectrum. This proves that MJI using a FPI can superimpose high-frequency spectral information into low-frequency spectral information, and high-frequency information can be obtained using the inversion algorithm.

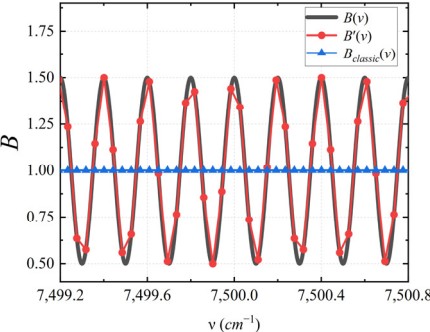

**Figure 8.** The result of the super-resolution spectrum when the input spectrum is a cosine function.

When the input spectrum is a Gaussian function, as shown in Figure 9a, the spectral resolution of $B'(v)$ is higher, and the value is more accurate than that of $B_{classic}(v)$. Through Gaussian fitting, the FWHM of $B'(v)$ and $B_{classic}(v)$ can be obtained as 0.101 cm$^{-1}$ and 0.197 cm$^{-1}$, respectively, as shown in Figure 9b, which also proves that the spectral resolution is improved by a factor of two. This extremely narrow Gaussian function often appears in the form of an absorption peak of a certain gaseous substance in the spectrum. Thus, information, such as the concentration of this gaseous substance, could thus be obtained by the inversion and calculation of the size of the absorption peak. In this application, the resolution and accuracy of spectral absorption peak measurements have a crucial impact on the accuracy of the inversion results.

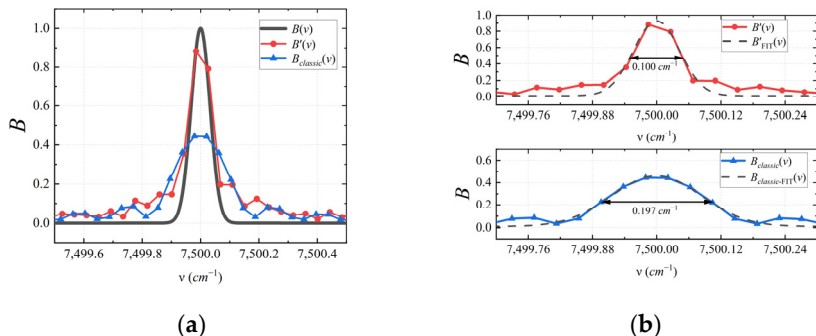

(**a**)                                      (**b**)

**Figure 9.** (**a**) The result of the super-resolution spectrum when the input spectrum is a Gaussian function; (**b**) FWHM of $B'(v)$ and $B_{classic}(v)$, where $B'_{FIT}(v)$ and $B_{classic\text{-}FIT}(v)$ are the Gaussian-fitting functions of $B'(v)$ and $B_{classic}(v)$, respectively.

When the input spectrum is chosen as spectral data, the result $I'_0(\Delta)$, compared to $I_0(\Delta)$, is as shown in Figure 10a. In Figure 10b–d, $I'_0(\Delta)$ and $I_0(\Delta)$ are amplified and compared at the low-, middle-, and high-frequency parts of $\Delta \in [L, 2L]$, respectively. It can be seen that the main, overall waveforms of the two interferograms are relatively consistent. Nonetheless, certain errors are still observable, which is associated with the simplification of the model in Equation (9).

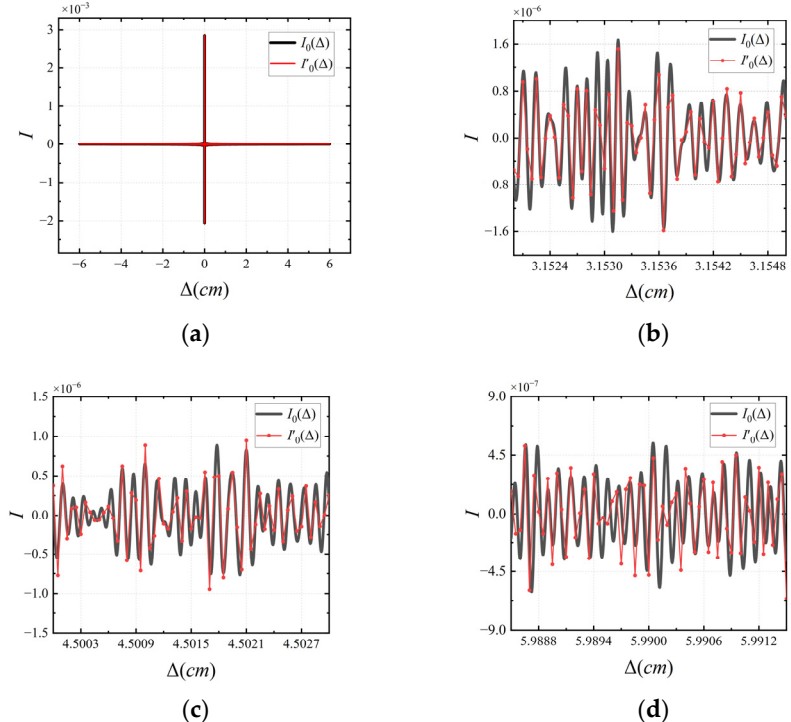

**Figure 10.** (**a**) The interferogram when the input spectrum is the spectral data; (**b–d**) the partial magnification comparison of this interferogram.

Figure 11a shows a spectral comparison diagram and Figure 11b–d presents partially enlarged diagrams of the case when the input spectrum is spectral data. It can be seen that the spectral resolution of $B'(v)$ is significantly improved compared to that of $B_{classic}(v)$, and many spectral details are preserved.

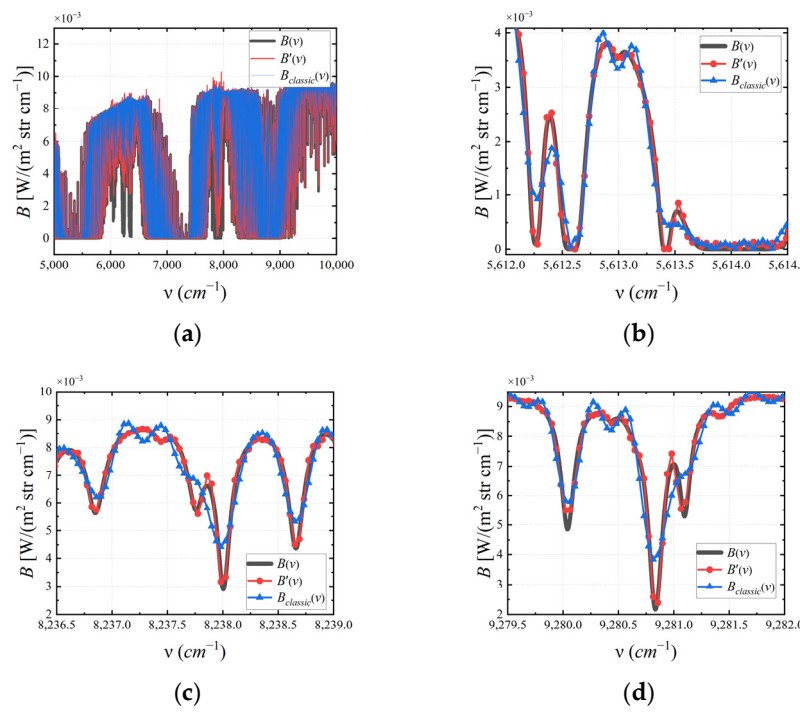

**Figure 11.** (**a**) The result of the super-resolution spectrum when the input spectrum is spectral data; (**b–d**) the partial magnification comparison of this spectrum.

### 4.3. Evaluation and Analysis

To evaluate the restoration accuracy, the error of the spectrum was quantified using the following formula [18]:

$$Error(\mathrm{dB}) = 10\lg\left[\frac{\sum\left[B'(v) - B(v)\right]^2}{\sum\left[B(v)\right]^2}\right] \tag{11}$$

As calculated by Equation (11), when the input spectrum is spectrum data, the spectral error of $B_{classic}(v)$ is −27.80 dB, whereas that of $B'(v)$ is −33.42 dB. The reduction in this part of the error is mainly due to the improvement in detail. The reflectivity $R$ determines the coefficient $a_m$, which in turn determines the linear equation system of Equation (10). Therefore, $R$ has a significant influence on the algorithm. The $T_{FPI}$ at different $Rs$ is shown in Figure 12a. When $R$ is determined, $\{a_m\}$ is a proportional sequence, and its relationship to $R$ is shown in Figure 12b. With the increase in $R$, the error caused by Equation (9) for model simplification becomes more significant. Figure 12c shows the errors under different reflectance values calculated using Equation (11).

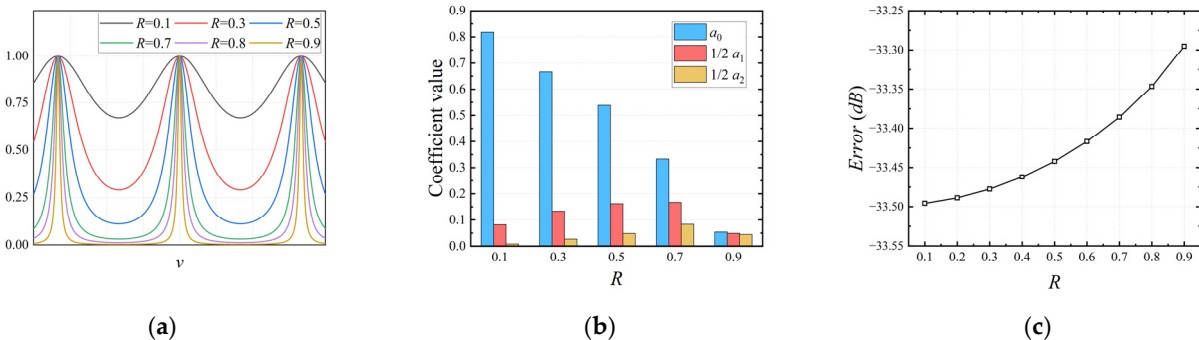

(**a**)  (**b**)  (**c**)

**Figure 12.** (**a**) The $T_{FPI}$ at different $Rs$; (**b**) the relationship between the reflectivity $R$ and the coefficient $a_m$; (**c**) the curve of Error vs. the reflectivity $R$.

In fact, the most critical step in the design of super-resolution systems is to establish a linear relationship between $I_1(\Delta)$ and $I_2(\Delta)$, $\Delta \in [-L, L]$ with respect to $I_0(\Delta)$, $\Delta \in [-2L, 2L]$, similar to that presented in Equation (10):

$$\begin{bmatrix} I_1 \\ I_2 \end{bmatrix} = A \cdot I_0 \tag{12}$$

where the inverse matrix $A$ is a large, sparse matrix, and $A$ must have a high numerical stability. We use the 2-Norm condition number $\kappa$ to evaluate the numerical stability of $A$. The reflectance $R$ of the FPI determines the coefficients of the inversion matrix $A$, and the interval $d$ determines its structure. Therefore, $R$ and $d$ together determine the stability of $A$. We calculated $\kappa$ for different values of $R$ and $d$. The calculation results show that when $2nd = 2L$ or $L$, $\kappa$ is small; that is, it has a high numerical stability, as shown in Figure 13. Under this condition, an $R$ that is too large or too small will increase $\kappa$. Nonetheless, $\kappa$ can be maintained at 60% in the range of 20% to 80%, which is excellent numerical stability. Compared to the resampling method, our proposed technique is more relaxed in terms of the reflectance requirement of the FPI. This advantage makes the technique more suitable for the enhancement of the spectral resolution for large-aperture or wide-spectral-range interferometric spectrometers with lower FPI preparation costs.

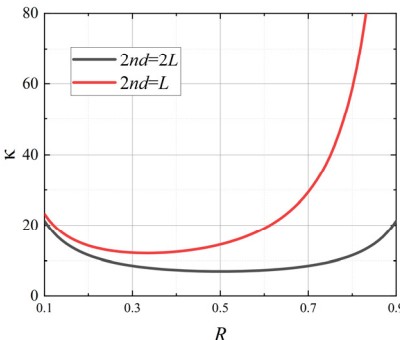

**Figure 13.** Relationship between the condition number κ and reflectance *R* at $2nd = 2L$ and $2nd = L$.

For a static FPI, the condition of $2nd = 2L$ (or $L$) is relatively easy to implement, and it has a higher stability without changing the interval *d*, as compared to the resampling method. If this condition cannot be satisfied, it is necessary to ensure that the interferogram is recoverable by sacrificing a part of the spectral resolution. For example, when $nd = 0.9L$, $2nd = 2 \times (0.9L)$ is satisfied, so $I_0(\Delta)$, $\Delta \in [-1.8L, 1.8L]$ can be obtained based on the inversion of $I_1(\Delta)$, $I_2(\Delta)$, $\Delta \in [-0.9L, 0.9L]$, where the spectral resolution is enhanced by 1.8, instead of 2, times. Or, when $nd = 0.4L$, $2nd = 0.8L$ is satisfied, and $I_0(\Delta)$, $\Delta \in [-1.6L, 1.6L]$ can be obtained based on the inversion of $I_1(\Delta)$, $I_2(\Delta)$, $\Delta \in [-0.8L, 0.8L]$, and the spectral resolution is enhanced by 1.6 times. Therefore, interval *d* determines the ability of the FPI to improve the spectral resolution. Based on this approach, *d* has a wide range of values and provides greater flexibility in the design of the FPI.

Based on the above analysis, the maximum OPD *L* of the Michelson interferometer and the interval *d* of the FPI jointly determine the spectral resolution, and the reflectivity *R* of the FPI affects the calculation accuracy.

### 4.4. Comparison and Analysis

In previous studies, the resampling super-resolution hyperspectral imager (RS-HI) was proposed [21–25]. The principle is to add a FPI to perform more detailed sampling within the spectral response of the imaging spectrometer and thus achieve spectral super-resolution. The technique is highly versatile and can be applied to both dispersive imaging spectrometers based on prisms or gratings, as well as interferometric imaging spectrometers [21–25]. The system structure of the RS-HI applied to the interferometric imaging spectrometer is close to that of Figure 2 [21,23]. However, it uses a tunable FPI, and there are major differences in the principle, system parameters, and applications. Therefore, this section will compare and analyze the MJI-HI and RS-HI.

The principle of the RS-HI is as shown in Figure 13 [21,23]. The multiple transmission peaks of the FPI can be used as narrow-band filters to form multiple narrow-band spectra, and Figure 13a,d,h indicates the transmission spectra at different intervals *d*, respectively. The obtained spectra from the interferometric spectrometer are the impulse responses of these narrow-band spectra, as shaded in Figure 13d,e,i. Integration of the $m_{th}$-level impulse response yields the energy $S_m$ of the narrow-band spectrum, as shown in Figure 13c,f,j. The super-resolution spectrum is obtained by changing the interval *d* several times and rearranging them according to the central wavelength of the narrow-band spectrum, as shown in Figure 13k. Based on this principle, the RS-HI can more easily achieve high-magnification spectral super-resolution, which is its greatest advantage compared to the MJI-HI. In addition, the RS-HI directly acquires narrow-band spectra between spectral sampling intervals, which is more suitable for dispersive imaging spectrometers, as based on the principle. This will bring great convenience to installation and calibration, etc. MJI-HI mainly processes and calculates the spectrum of a spectrum and is more suitable for interferometric imaging spectrometers.

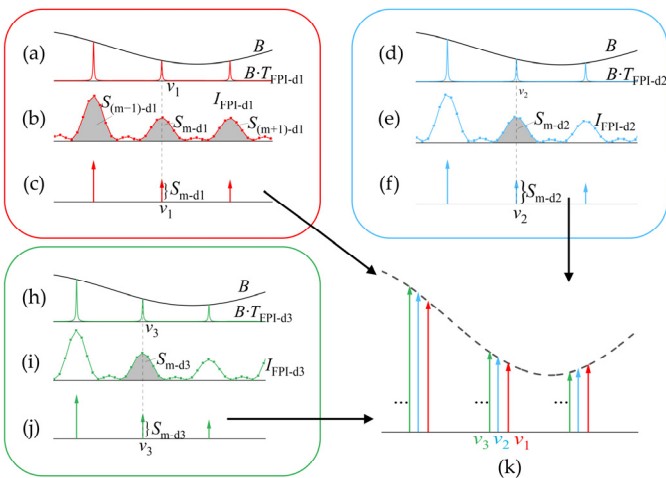

**Figure 14.** Schematic diagram of the RS-HI super-resolution principle: (**a**,**d**,**h**) incident spectrum *B* and spectrum $B \cdot T_{FPI-d}$ obtained by passing through the FPI with intervals d1, d2, and d3, respectively; (**b**,**e**,**i**) low-resolution spectrum obtained by interferometric spectrometer; (**c**,**f**,**j**) the energy S of the narrow-band spectrum is obtained based on the integration of the low-resolution spectrum; (**k**) the super-resolution spectrum is obtained by combining all of the S according to the central wavelength of the narrow-band spectrum.

The transmission spectrum of a FPI vs. reflectance is shown in Figure 12a. The interference intensity is usually defined as $K = (T_{FPI-MAX} - T_{FPI-MIN})/(T_{FPI-MAX} + T_{FPI-MIN})$, where $T_{FPI-MAX}$ is the maximum value of $T_{FPI}(v)$, and $T_{FPI-MIN}$ is the maximum value of $T_{FPI}(v)$. When *K* is large enough, $T_{FPI}(v)$ can be viewed as a comb-like transmission spectrum consisting of multiple transmission peaks. Additionally, *K* is only related to the fineness *F* of FPI, which depends on the reflectivity *R*. The actual engineering will lead to degradation due to the optical and mechanical tolerances of a FPI. The most fundamental problem of the RS-HI is that the FPI needs to produce transmission peaks with a very narrow width and a stable central wavelength. According to Figure 12a, when the reflectivity can reach more than about 90% and the FPI has a small enough optical tolerance, *K* will be high enough to form transmission peaks. According to the results in Section 4.3, the *R* of the FPI in the MJI-HI should be in the range of 20~80%. Excessive reflectance will lead to a greater processing difficulty and a higher cost for the actual design, and it will also reduce the luminous flux significantly, and thus the signal-to-noise ratio, which will weaken the advantage of the multichannel interferometric spectrometer.

In addition, the spacing *d* of the FPI in the RS-HI is on the order of mm or cm, which is close to the order of the maximum OPD *L* of a Michelson interferometer. additionally, the scanning step of *d* during the measurement is approximately on the order of nm [21,23]. To obtain super-resolution spectra, the sampling points need to be aligned according to the central wavelength, which mainly relies on theoretical calculations. The central wavelength also varies with the field-of-view angle. To ensure the accurate wavelength of the center of the transmission peak at all levels of the entire field of view, the tuning accuracy of the FPI needs to be controlled on the order of nm. This kind of FPI, with a large interval, a small tuning range, and a very high level of accuracy, is difficult to manufacture and process. A Michelson interferometer requires strict system accuracy, and two interferometers with moving parts will make the whole system less stable. In contrast, the MJI-HI uses a static FPI, which ensures that the system stability is not greatly affected and reduces both the processing difficulty and the cost of the FPI.

As shown in Figure 14b, the RS-HI part of the error comes from the sidelobes of the impulse response of the interferometric spectrometer, and the surrounding sampling points will integrate this part of the sidelobes and thus cause the error, $S_{(m-1)-d1}$, and $S_{m-d1}$ and $S_{(m+1)-d1}$, as shown in the figure, will affect each other. This can lead to pseudo-

spectra in parts of the spectrum, with large undulations, resulting in a change in the shape of the spectrum. This problem can be alleviated by apodization and increasing the FSR. However, increasing the FSR leads to an increase in the width of the FPI transmission peak, and a further increase in *R* is required to ensure the same spectral resolution. Additionally, increasing the FSR also means that more spectral data between the two sampling points of the FPI will not be used, i.e., the data points outside the shading in Figure 14b, and the data utilization will be further reduced. Therefore, in the FPI spacer design, a trade-off between accuracy and reflectance and data utilization is required to ensure the spectral resolution. In contrast, the RS-HI based on a dispersive imaging spectrometer, the response function of which can be regarded as a rectangular function, does not produce similar errors without considering diffraction.

The principle behind the MJI-HI's super-resolution can be equated to expand the *L*. Therefore, the super-resolution spectrum is still uniformly sampled, which is more conducive to the processing and application of spectral data. Moreover, the MJI-HI can obtain twice the super-resolution spectrum using two interferograms $I_1$ and $I_2$, in which there is no data waste. The RS-HI, however, has the problem of non-uniform sampling. This is due to the fact that the tunable FPI changes the interval *d*; i.e., it changes the FSR, rather than keeping the FSR constant over all of the transmission peak shifts. As shown, Figure 15a–c corresponds to the super-resolution spectra sampled at the long-, medium-, and short-wavelength positions in the measurement spectral range, respectively. In Figure 15b, the FPI sampling range at the mid-wavelength position covers exactly one FSR, while a larger region in the long-wavelength position exists and is not sampled, as in the red-boxed region in Figure 15a. This would result in multiple such regions within the spectrum not being acquired. In addition, sampling more than one FSR at short-wavelength locations can lead to the repeated sampling of the same spectrum, producing meaningless duplicate data. So, the tuning range of the FPI needs to be traded off in terms of spectral coverage and data utilization. Dispersive imaging spectrometers have a much smaller spectral range and lower spectral resolution, and in this regard, the super-resolution method of the RS-HI is more suitable for dispersive imaging spectrometers.

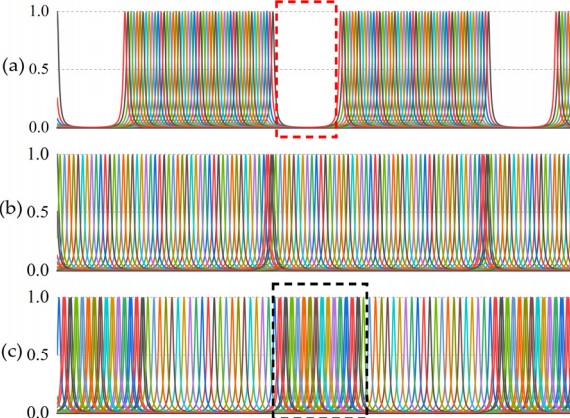

**Figure 15.** Sampling of a tunable FPI in (**a**) long-wavelength; (**b**) mid-wavelength; (**c**) short-wavelength ranges.

We used the atmospheric absorption spectrum in Section 4.1 as the input spectra. The results of the super-resolution of the Michelson interferometer spectra for different *Ls* were simulated when the RS-HI spectral resolution was consistent with that simulated by the MJI-HI discussed in Section 4.1. The interval *d* of the FPI in the simulation is 0.2 cm, and the interval scan step is 14 nm, covering the mid-wavelength position FSR is at least 50 scan steps, and for covering the whole spectrum, the number of scan steps is 70. Trigonometric toe-shearing is used to reduce the influence between adjacent sampling points,

ignore the tolerance, and simulate the RS-HI under ideal conditions. The error of the super-resolution spectrum is calculated according to Equation (11). Assuming that $\alpha$-fold spectral super-resolution is achieved using $\beta$ interferograms, the data volume required is defined as $\beta \times L/l$, and the data utilization rate is $\alpha/\beta$. The MJI-HI is a two-fold spectral super-resolution using two interferograms, so the data volume is $1.2 \times 10^5$, and the data usage rate is 100%. The maximum optical range $L$ of the Michelson interferometer used is 0.6~3.0 cm, respectively. The simulation results are shown in Table 1. The results show that the error decreases rapidly in the range of $L$ = 0.6~1.2 cm with the increase of $L$. This is due to the fact that the impulse-response width of the Michelson interferometer decreases, and the mutual influence between adjacent sampling points is reduced. The small variation in errors in the range of $L$ = 1.2~3 cm indicates that, to achieve the same spectral resolution, an interferometric spectrometer with a smaller optical range could be used, which would have less of an impact on the accuracy, it would significantly reduce the volume of the data to be collected, and it would improve the data utilization. In the same case of a 2-fold super-resolution for the $L$ = 3.0 cm spectrum, it is slightly higher than the error of the MJI-HI discussed above (−33.42 dB). However, with this same spectral resolution and close error, the RS-HI collects 35 times more data than the MJI-HI, and the data utilization is lower. This means that the RS-HI is more suitable for the high magnification of spectral super-resolution, and the MJI-HI is more suitable for the low magnification of spectral super-resolution. By increasing the spectral resolution of the FPI to 0.0217 cm$^{-1}$, the error of the RS-HI is further reduced. The spectral super-resolution was performed for the same Michelson interferometer, and the spectral resolution was kept around 0.0217 cm$^{-1}$ at different intervals $d$ by adjusting $R$. The results show that its error rises dramatically as $d$ approaches $L$, and the opposite requires a higher reflectivity. This also confirms the need to balance the trade-off between accuracy and $R$ when designing $d$ with guaranteed spectral resolution.

**Table 1.** Parameters and results of the RS-HI simulation.

| FPI-$d$ (cm) | FPI-R | FPI-FWHM (cm$^{-1}$) | $L$ (cm) | Resolution Improvement Multiplier | Data Volume | Data Utilization Rate | Error (dB) |
|---|---|---|---|---|---|---|---|
| 0.2 | 95.00% | 0.0816 | 0.6 | 10.2 | $8.4 \times 10^5$ | 14.57% | −27.668 |
| | | | 0.7 | 8.7 | $9.8 \times 10^5$ | 12.44% | −28.619 |
| | | | 0.8 | 7.6 | $1.12 \times 10^6$ | 10.86% | −29.650 |
| | | | 1.2 | 5.1 | $1.68 \times 10^6$ | 7.29% | −31.599 |
| | | | 1.8 | 3.4 | $2.52 \times 10^6$ | 5.06% | −32.763 |
| | | | 2.4 | 2.5 | $3.36 \times 10^6$ | 3.57% | −33.426 |
| | | | 3.0 | 2.0 | $4.2 \times 10^6$ | 2.86% | −33.936 |
| 1 | 93.40% | | | | | | −33.232 |
| 0.75 | 95.00% | 0.0217 | 3.0 | 7.7 | $4.2 \times 10^6$ | 11.0% | −39.825 |
| 0.5 | 96.65% | | | | | | −41.6513 |

In summary, compared with the RS-HI, the MJI-HI has greater advantages in FPI parameter requirements, system stability, optical-component processing difficulty, and cost. Based on our analysis of the super-resolution principle, spectral sampling vacancy, low data utilization, and interference between response functions, the RS-HI is more suitable as a dispersive imaging spectrometer, and the MJI-HI is more suitable as an interferometric imaging spectrometer. According to the simulation results, the RS-HI is more suitable for high-magnification spectral super-resolution, and the MJI-HI is more suitable for low-magnification spectral super-resolution. Therefore, the MJI-HI is suitable for low-magnification interferometric spectroscopy super-resolution and has the advantages of high data utilization, high stability, and low cost.

## 5. Conclusions

In this study, we investigated the method of adding a FPI to an interferometric spectroscopic imager to achieve spectral super-resolution. Conventional hyperspectral imaging systems cannot obtain a continuous spectrum and will eventually lose detailed high-frequency information outside the cutoff frequency. These optical systems can be regarded as low-pass systems of spectral dimension. Here, we showed that the addition of a FPI can achieve a breakthrough in the cut-off frequency of the low-pass system and improve the spectral resolution. The role of the FPI is that the transmission spectrum can be used as a modulated wave to achieve spectrum mixing, and the double-beam interferometer can obtain an interferogram of the spectrum after mixing. It can be viewed as a linear combination of the multiple displacement components of the original interferogram that moves high-frequency interferometric information beyond the maximum OPD into the observable range. Therefore, high-frequency interference information is calculated by constructing a system of linear equations, which is equivalent to increasing the maximal OPD and achieving double-spectral super-resolution. Theoretically, spectral resolution can be improved multiple times based on this idea. Compared to the resampling method, the proposed approach in this paper requires a low level of reflectance for the FPI, no interval change, and less difficulty in spectral reconstruction, and it is more suitable for improving the spectral resolution of interferometric imaging spectrometers. We have introduced the Michelson interferometer as an example, but the proposed method could also be applied to other types of double-beam interferometers. We simulated the method using a high-frequency cosine function, a Gaussian function, and spectral data as the input spectrum. The simulation results show that this method successfully achieves the high-precision recovery of spectral high-frequency information, and that the spectral resolution is significantly improved. The interval and reflectance of the FPI are two important optical parameters in the proposed method. We analyzed the effects of the two parameters on the proposed method, where the interval size determines the magnitude of the spectral resolution, and the reflectance affects the accuracy of the interferogram recovery. Finally, we performed a comparative analysis with related studies. The results show that the MJI-HI is more suitable for the spectral super-resolution of low-resolution interferometric spectrometers, and it has the advantages of high stability, high data utilization, and low cost.

The method proposed in this study to improve spectral resolution using the FPI does not provide a complete analysis of the FPI or the optical tolerances of the entire optical system. Therefore, tolerance analysis and subsequent experimental verification will be the main focuses of our subsequent research. In addition, this method uses only one displacement component, whereas multiple displacement components could be used to achieve multiple improvements in spectral resolution. We will conduct related research on these issues in the future.

**Author Contributions:** Conceptualization, Q.L., Y.L., and Y.Z.; methodology, Y.Z. and Q.L.; software, Y.Z.; validation, Y.Z. and Q.L.; formal analysis, Y.Z., Y.T., and P.H.; investigation, Y.Z., B.Z., X.S., Y.Y., and Y.B.; writing—original draft preparation, Y.Z., Y.L., and Q.L.; writing—review and editing, Y.Z., Q.L., and Y.L.; project administration, Q.L.; funding acquisition, Y.L. All authors have read and agreed to the published version of the manuscript.

**Funding:** This research was funded by the Key Program Project of Science and Technology Innovation of the Chinese Academy of Sciences (no. KGFZD-135-20-03-02), the Science Technology Foundation Strengthening Field Fund Project (no.2020-JCJQ-JJ-492), and the Strategic Priority Research Program of the Chinese Academy of Sciences (no. XDA28050101).

**Institutional Review Board Statement:** Not applicable.

**Informed Consent Statement:** Not applicable.

**Data Availability Statement:** Not applicable.

**Conflicts of Interest:** The authors declare no conflicts of interest.

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
