# Peer review of "Super-Resolution Multicomponent Joint-Interferometric Fabry–Perot-Based Technique"

_applsci, doi:10.3390/app13021012_

Round 1
Reviewer 1 Report
(1) Abstract: Clearly mention that your research work is a simulation work. Key achievements (quantitative analysis) of your research works can be included.
(2) Introduction: Clearly mention that your research work is a simulation work. Quantitative analysis can be included in Page 2/3 (Lines 91-103).
(3) 4. Experimental Simulation and Results Discussion
The section title can be changed to "Results and Discussion."
4.1. Experimental simulation conditions
The sub-section title can be changed to "Simulation conditions." Also change the necessary language throughout the manuscript. Mention the type of simulator used for the research.
4.2. Experimental simulation results
The sub-section title can be changed to "Simulation Results."
Comparison of your research works with other reported research works is required.
Reviewer 2 Report
The manuscript of Yu Zhang et al. concerns the novel theoretical approach for super-resolution spectral imaging. It is based on the core concept of structured illumination microscopy, enabling super-resolution imaging below diffraction limits in the spatial domain. However, in a recent study, the same concept was used to increase the spectral resolution. It is an interesting approach. The organization of the manuscript is appropriate; however, sections 2 and 3 concerning the used methods are too detailed in my opinion. The authors used appropriate and recent references. The language used is understandable. However, before publication, some additional explanation and corrections are necessary.
Major comments:
1) The main weakness of this manuscript is the lack of experimental verification of the assumptions made to confirm the results of the numerical analysis. This makes this manuscript more a case report type than full article. The authors should consider the extension of the manuscript by introducing some experimental results enabling the verification and validation of the simulation results and presented theoretical considerations.
2) According to Section 4.2, if the ideal interferogram and the super-resolution interferogram are the same as was stated in lines 269-271, therefore, what is the source of differences between the input spectrum B(v) and super-resolution spectrum B’(v), if they are the inverse Fourier transform of the same input functions as it was indicated by the authors?
3) The representation of the results in Fig.7 is somewhat misleading, the signal I' is the envelope of the input signal? Is this just due to the overlap of the signal and the different thickness of the lines used on this figure?
4) The same situation is in case of Fig.11a, the authors should consider different, more representative and understanding for the readers way of demonstration of results.
Minor comments:
1) Among the authors, only one of them (second author) was indicated as a person contributing equally to this work. The second person is the first author?
2) Lack of spaces: e.g. line 218, lines 298-299 etc.
3) The subfigures of one figure should be located on the same page.
Reviewer 3 Report
The manuscript is interisting but need some improves:
More results and a comparation with other configuration.
Applications for this system. the authors resfers in conclusions but is necessary before.
The equations needs references.
I recommend a major reviews.
Round 2
Reviewer 2 Report
The explanations provided by the authors and the corrections made to the manuscript are satisfactory. I hope that the authors succeed in constructing a measuring system and successfully applying the proposed methods to experimentally verify the assumptions declared here. In my opinion, the manuscript is ready for publication in its present form.
Reviewer 3 Report
The authors proprose a new version of manuscritp. In this version the authors clarify the question in the last review. i recommend publish.